# *RYK* Gene Expression Associated with Drug Response Variation of Temozolomide and Clinical Outcomes in Glioma Patients

**DOI:** 10.3390/ph16050726

**Published:** 2023-05-10

**Authors:** Ricardo D. Gonzalez, George W. Small, Adrian J. Green, Farida S. Akhtari, Tammy M. Havener, Julia C. F. Quintanilha, Amber B. Cipriani, David M. Reif, Howard L. McLeod, Alison A. Motsinger-Reif, Tim Wiltshire

**Affiliations:** 1Division of Pharmacotherapy and Experimental Therapeutics, UNC Eshelman School of Pharmacy, University of North Carolina at Chapel Hill, Chapel Hill, NC 27599, USA; rgonzalez1@unc.edu (R.D.G.); george_small@med.unc.edu (G.W.S.); amber.cipriani@unchealth.unc.edu (A.B.C.); 2Center for Pharmacogenomics and Individualized Therapy, University of North Carolina at Chapel Hill, Chapel Hill, NC 27599, USA; 3Department of Biological Sciences, North Carolina State University, Raleigh, NC 27606, USA; ajgreen4@ncsu.edu; 4Bioinformatics Research Center, North Carolina State University, Raleigh, NC 27606, USA; 5Biostatistics and Computational Biology Branch, Division of Intramural Research, National Institute of Environmental Health Sciences, Research Triangle Park, NC 27709, USA; farida.akhtari@nih.gov (F.S.A.); alison.motsinger-reif@nih.gov (A.A.M.-R.); 6Structural Genomics Consortium and Division of Chemical Biology and Medicinal Chemistry, Eshelman School of Pharmacy, University of North Carolina at Chapel Hill, Chapel Hill, NC 27599, USA; thavener@unc.edu; 7Clinical Development, Foundation Medicine, Boston, MA 02115, USA; 8Predictive Toxicology Branch, Division of Translational Toxicology, National Institute of Environmental Health Sciences, Research Triangle Park, Durham, NC 27709, USA; david.reif@nih.gov; 9Center for Precision Medicine and Functional Genomics, Utah Tech University, St. George, UT 84770, USA

**Keywords:** GWAS, chemotherapy, sensitivity, resistance, glioma, drug response, temozolomide, knockdown, overexpression, pharmacogenomics, gene expression, *RYK*

## Abstract

Temozolomide (TMZ) chemotherapy is an important tool in the treatment of glioma brain tumors. However, variable patient response and chemo-resistance remain exceptionally challenging. Our previous genome-wide association study (GWAS) identified a suggestively significant association of SNP rs4470517 in the *RYK* (receptor-like kinase) gene with TMZ drug response. Functional validation of *RYK* using lymphocytes and glioma cell lines resulted in gene expression analysis indicating differences in expression status between genotypes of the cell lines and TMZ dose response. We conducted univariate and multivariate Cox regression analyses using publicly available TCGA and GEO datasets to investigate the impact of *RYK* gene expression status on glioma patient overall (OS) and progression-free survival (PFS). Our results indicated that in *IDH* mutant gliomas, *RYK* expression and tumor grade were significant predictors of survival. In *IDH* wildtype glioblastomas (GBM), *MGMT* status was the only significant predictor. Despite this result, we revealed a potential benefit of *RYK* expression in *IDH* wildtype GBM patients. We found that a combination of *RYK* expression and *MGMT* status could serve as an additional biomarker for improved survival. Overall, our findings suggest that *RYK* expression may serve as an important prognostic or predictor of TMZ response and survival for glioma patients.

## 1. Introduction

Gliomas are the most common (~75%) form of malignant brain and central nervous system tumors (CNS) [1,2,3,4,5]. The World Health Organization (WHO) released new guidelines for the classification and diagnosis of gliomas in 2021 (CNS5). These guidelines showcase significant changes by placing a greater emphasis on the utilization of molecular diagnostics in classifying CNS tumors, while still retaining established approaches to tumor characterization such as histology and immunohistochemistry [4].

Important molecular features, such as isocitrate dehydrogenases (*IDH*) mutations, which require testing for the workup of all gliomas, and O-6-Methylguanine-DNA methyltransferase (*MGMT*) promoter methylation are important clinical markers in adult-type infiltrating tumors, affecting patient outcomes [6,7,8,9,10]. In the current edition of the WHO CNS5, the classification of gliomas has been restructured, with six families now identified, including adult-type diffuse gliomas, pediatric-type low-grade and high-grade diffuse gliomas, circumscribed astrocytic gliomas, glioneuronal and neuronal tumors, and ependymomas. Compared to the 2016 WHO classification, which divided adult-type diffuse gliomas into 15 entities, the CNS5 has simplified this to three types: astrocytoma, *IDH*-mutant; oligodendroglioma, *IDH*-mutant and 1p/19q-codeleted; and glioblastoma, *IDH*-wildtype (GBM). Other significant changes include the systematic categorization of pediatric-type diffuse gliomas based on their established genetic alterations [4]. GBMs prioritize the assessment and inclusion of *MGMT* promoter methylation as an essential component of molecular diagnostics and treatment for all high-grade gliomas (grades 3 and 4) where studies have reported that up to 60% of GBMs cases exhibit *MGMT* promoter methylation [4,9,11,12,13]. These markers have opened up opportunities for pharmacogenomics to contribute to our understanding of how various tumor subtypes and genetic variations can influence an individual’s response to TMZ treatment, and for developing a more refined understanding of mechanisms underlying TMZ resistance. This classification system allows for a more precise diagnosis and treatment plan for patients with gliomas.

First-line therapy for glioma typically involves cytoreductive surgery and gross total surgical resection, when feasible, followed by radiation therapy (RT) and concurrent and/or adjuvant temozolomide (TMZ) chemotherapy, particularly for high-grade glioma [14]. Many patients become tolerant or resistant to treatment as the median survival rate is only about 15 months after first diagnosis, and the 5-year survival rates remain poor (<10%) despite advances in surgical and medical neuro-oncology [15,16]. Apart from TMZ, several other drugs are utilized in glioma treatment, such as bevacizumab, carmustine, lomustine, procarbazine, and vincristine [13,17].

Pharmacogenomics is becoming an important tool in predicting the response for successful advancement of glioma stratification and effective personalized treatments in precision medicine and oncology [18,19].

In vitro models of human gliomas have the potential to improve our understanding of glioma biology and novel therapeutic strategies [20]. High throughput drug screens utilizing cell line models have assessed the sensitivity and resistance of different compounds and identified both germline and somatic variants that influence drug response [21,22,23]. The 1000 Genomes Project collection is a valuable resource for studying the impact of diverse global human genetics on drug response and disease. Immortalized B lymphocytes (LCLs) from individuals are available with extensive genomic characterization, providing an opportunity to develop drug cytotoxicity profiles. This allows for genome-wide association studies (GWAS) and gene expression studies in large multi-ethnic cohorts [24,25]. Cancer cell line systems also serve as useful in vitro models for a specific tumor type to study cancer biology and treatment responses [20,26,27].

In this study, we have explored a potential genetic factor influencing response to TMZ from a high-throughput assay using LCLs and glioma cell lines [21,28]. TMZ dose response was investigated due to its widespread use in the treatment of gliomas and its documented association with chemoresistance, which remains a significant challenge in glioma treatment [18]. Given the observation of a significant association of *RYK*-related variants from previous analyses, we conducted follow-up experiments to validate the role of *RYK* in TMZ drug response.

## 2. Results

### 2.1. GWAS Analysis

A previous high-throughput dose-response screen and GWAS of 44 anticancer drugs using 1000 Genome LCLs was performed by Akhtari et al. [21], which revealed 10 unique SNP associations at genome wide significance and 40 unique SNP associations at suggestive significance associated with drug response across 21 different anticancer agents. Analysis revealed a previously unreported association between TMZ response and genetic variants, which is presented on a Manhattan plot in Figure 1a. This finding has not been previously reported on individually. The peak centers on chromosome 3 and has two rs4470517 (*p*-value = 7.08 × 10^−7^) and rs4854617 (*p*-value = 3.35 × 10^−7^) SNPs that surpass suggestive significance. Figure 1b shows the locus zoom plot of the regional genes within this peak for association with TMZ. rs4470517 and rs4854617 are located within the *RYK* gene that encodes for a transmembrane atypical member of the receptor tyrosine kinase (RTK) family. rs4470517 and rs4854617 are intronic SNPs located between exons 13–14 and 14–15, respectively. In both SNPs, the alternate allele A is the major allele in the population. In 1000 Genome data, rs4470517 among a European population have a minor allele frequency (MAF) of G = 0.317115, whereas an Eastern Asian population have the lowest MAF of G = 0.0019 (Appendix A). Since both SNPs were in a high degree of linkage disequilibrium (R^2^: 0.8968) we focused our functional assessment on rs4470517 as it was predicted to have a deleterious phenotype (Appendix A).

### 2.2. Effect of RYK SNP rs4470517 on TMZ Dose Response

To determine if this SNP is functionally relevant, we investigated the association of *RYK* gene expression levels and assessed how genotypes are related to TMZ dose response. We show that SNP rs4470517 is significantly associated with TMZ dose response. Figure 2a shows LCL’s dose response profiles stratified by genotype. LCLs from individuals with the AA genotype at rs4470517 have lower cell viability when exposed to all concentrations tested compared to those with the GG genotype. Similarly, Figure 2b shows that the AA genotype cells have higher area under the curve (AUC) response (1—viability), indicating a higher drug response and sensitivity to TMZ than the GG genotype cell lines (*p* = 0.01). To assess relevance in non-LCL cell lines, glioma cell line U138MG was found to have an AA genotype and had the lowest cell viability when treated with TMZ among the glioma cell lines (IC_50_ = 0.57 mM). Glioma cell lines U87MG and U373MG were found to have an AG genotype, and T98G was found to have a GG genotype and a higher cell viability (IC_50_ = 1.44 mM) (Figure 2c).

### 2.3. Effect of RYK SNP rs4470517 on RYK Gene Expression

We then investigated the association of *RYK* gene expression levels and TMZ dose response by genotype. We also examined the associations between baseline and TMZ-induced expression levels by RT-qPCR after 72 h. In the LCLs, there were statistically significant differences in baseline expression of the ENSG00000163785 *RYK* transcript and mRNA levels between the AA and GG genotyped cells (Figure 2d, *p* = 0.03). Baseline expression of *RYK* was found to be negatively correlated (*p* < 0.001) with cell response to TMZ (Appendix A). There were no significant differences between groups for drug-induced expression, although both groups express significantly more *RYK* when exposed to TMZ (Appendix A). We also examined TMZ-induced *RYK* gene expression profiles after 72 h by RT-qPCR in the glioma cell lines. In the U138MG cell line (AA), *RYK*-induced expression was not significantly changed or reduced (Appendix A), having at least one G allele as found in the U87MG, U373MG, and T98G cell lines that had increased *RYK* expression when treated at all TMZ concentrations.

### 2.4. RYK Overexpression Protects Cells against TMZ Treatment

To explore the role of *RYK* in regulation of TMZ response in glioma cells, we overexpressed *RYK* in a HEK293 cell line (AG) and a U138MG glioma cell line (AA) utilizing a lentiviral-based plasmid under a CMV promoter for transient transfection. We hypothesized that overexpressing *RYK* would result in comparable increased resistant phenotype as with the GG genotyped cells, with increases in viability and gene expression seen in the LCLs. *RYK* overexpressed HEK293 and U138MG cells have an increase in gene expression and protein levels confirmed by qPCR and Western blotting (Appendix A). In *RYK* overexpressed HEK293 cells, the increase in *RYK* led to a significant increase in cell viability; IC_50_ = 1.74 mM compared to the HEK293 control cells, IC_50_ = 1 mM (Figure 3a). *RYK* overexpression in U138MG cells also had an increase in cell viability compared to the U138MG control cells (Appendix A).

### 2.5. RYK siRNA Knockdown Sensitizes Cells toward TMZ

To continue exploring the role of *RYK* in the regulation of TMZ response in glioma cells, we produced *RYK* knockdowns via siRNAs. Having previously observed that TMZ treatment reduced *RYK* expression in the cell lines with the AA genotype, we hypothesized that *RYK* knockdowns would result in a comparably, more sensitive phenotype as those with the AA genotype. *RYK* knockdown experiments were performed in HEK293 (AG) and T98G (GG) cells to demonstrate the influence that *RYK* expression has on sensitizing TMZ drug response. HEK293 and T98G *RYK* knockdown cells have decreased gene expression and protein levels confirmed by qPCR and Western blotting (Figure 3d,f). HEK293 *RYK* knockdown led to a significant reduction in cell viability, IC_50_ of 0.74 mM compared to the HEK293 control cells (Figure 3b). *RYK* knockdown in T98G cells was less pronounced with a slight reduction in cell viability compared to the T98G controls (Appendix A).

### 2.6. RYK Expression Is Associated with Clinical Prognosis in Adult-Type Diffuse Glioma Patients

We utilized Kaplan–Meier curves and Cox regression analysis, focusing on the overall survival (OS) of adult-type diffuse glioma patients and their *RYK* gene expression levels from the Cancer Genome Atlas (TCGA) dataset [29]. To gain deeper insights into the molecular mechanisms driving survival outcomes in diffuse gliomas, we have categorized these tumors based on a common molecular feature used in diagnosis, namely, the *IDH* gene mutation status. We grouped astrocytomas and oligodendrogliomas together as *IDH* mutants, while glioblastomas are categorized as *IDH* wildtype. However, we acknowledge that each tumor type has its own distinct genetic and molecular features, and we have also taken into account the tumor type in our analysis to obtain a more comprehensive understanding of the factors influencing patient survival. To assess the prognostic significance of *RYK* expression status, we performed univariate Cox regression analysis. Furthermore, we conducted multivariate analysis to evaluate the impact of adult diffuse glioma-type, grade, *RYK* expression, and *MGMT* methylation on patient survival. Due to changes in glioma testing and classification guidelines in recent years, the TCGA and GEO databases may not always reflect current guidelines, resulting in outdated classifications and varying molecular information. To address this, we reviewed and revised data from the TCGA CNS and GEO databases wherever possible to align with the current CNS5 guidelines and excluded cases with missing data that were unclassifiable. Our revised TCGA dataset includes 135 cases of *IDH*-mutant astrocytoma (53 grade 2 and 82 grade 3), 166 cases of *IDH*-mutant oligodendroglioma (100 grade 2 and 66 grade 3), and 116 cases of *IDH* wildtype glioblastoma grade 4, all with *RYK* expression and survival outcomes available for analysis.

The univariate Cox regression analysis revealed that *RYK* expression status (high vs. low, hazards ratio (HR) = 0.46; 95% confidence interval (CI), 0.23–0.94, *p* = 0.033) (Appendix A) was a significant risk factor for OS in the *IDH* mutant TCGA glioma patient subgroup. After performing multivariate analyses, neoplasm histologic grade (G3 vs. G2, HR = 3.72; 95% CI, 2.02–6.83, *p* < 0.001) and *RYK* expression status (high vs. low, HR = 0.44; 95% CI, 0.22–0.91, *p* = 0.027) (Figure 4a) were significant risk factors for OS in this patient group. However, cancer type and *MGMT* methylation status were not significant risk factors, indicating that *RYK* expression remained as a potential independent prognostic factor in this patient group.

In contrast, in the GBM, *IDH* wildtype TCGA glioma patient subgroup, *RYK* expression status (Appendix A) was not a significant risk factor for OS. Furthermore, after multivariate analyses, the results showed that *RYK* expression and *MGMT* status were not significant risk factors for OS in TCGA GBM patients (Figure 4b). Kaplan–Meier curves and forest plots of univariate results are shown in Appendix A.

### 2.7. RYK Expression Status Associated with Clinical Prognosis in Adult-Type Diffuse Glioma Patients from Independent Trials

In addition to our analysis of the TCGA CNS dataset, we reanalyzed available data from two GEO datasets (GSE7696, consisting of 76 *IDH* wildtype grade 4 GBM cases, and GSE107850, consisting of 166 *IDH* mutant NOS (not otherwise specified) grade 2 cases) to investigate the impact of *RYK* gene expression and *MGMT* methylation on the progression-free survival (PFS) and OS of patients who received TMZ treatment. To this end, we conducted both univariate and multivariate Cox regression analyses. However, since the GSE107850 dataset had limited availability of *MGMT* methylation status data, our analysis may be subject to the influence of this factor.

In our univariate analysis, we observed a trend indicating that *RYK* expression was linked to improved survival outcomes in *IDH* mutant grade 2 gliomas cases. However, this association did not reach statistical significance as a risk factor for PFS (Figure 5a).

Our subsequent univariate analysis of *IDH* wildtype grade 4 GBM cases revealed that both low (vs. high) and medium (vs. high) *RYK* expression levels were significantly associated with improved OS (HR = 0.46; 95% CI, 0.25–0.87; *p* = 0.016, and HR = 0.52; 95% CI, 0.28–0.97; *p* < 0.04, respectively) (Appendix A). Furthermore, *MGMT* status was also identified as a significant risk factor for OS, unmethylated vs. methylated (HR = 4.3; 95% CI, 2.4–7.5, *p* < 0.001) (Appendix A).

After conducting multivariate analyses, we found that *MGMT* methylation, unmethylated vs. methylated (HR = 3.86; 95% CI, 2.15–7.0, *p* < 0.001) remained a significant risk factor for OS (Figure 5b). Notably, low and medium *RYK* expression levels were identified as significant risk factors for OS in GBM, *IDH* wildtype patients who were *MGMT* methylated (HR = 0.37; 95% CI, 0.15–0.92, *p* = 0.033 and HR = 0.36; 95% CI, 0.14–0.92, *p* = 0.032, respectively) (Appendix A). Kaplan–Meier curves and forest plots are available in Appendix A.

## 3. Discussion

Utilizing genetic information to guide and tailor treatment for individuals is a key focus of personalized medicine. Here we identify and validate a gene associated with TMZ response using data from in vitro screening of LCLs. As a result of the GWAS signal, we identified the SNP rs4470517 within the *RYK* locus as a suggestive genetic factor linked to TMZ response and examined a functional role for *RYK* as a potential marker of TMZ response.

Experiments showed that this association is likely due to the regulation of *RYK* gene expression by this SNP, or linked SNPs. We observed that cell lines with the AA variant have significantly decreased *RYK* expression compared to the GG genotype cell lines, which is associated with a more active drug response to TMZ. Specifically, the AA variant cell lines had a higher AUC in response to TMZ compared to the GG genotype cell lines, which showed a more resistant-like phenotype with a less active drug response to TMZ. We also found that knocking down *RYK* expression in cell lines with the GG or AG genotype, which are more resistant, resulted in a more sensitive phenotype and increased responsiveness to TMZ treatment. This suggests that *RYK* may be involved in the resistance mechanisms of these genotypes and reducing *RYK* expression could overcome this resistance and improve TMZ response. Our findings are consistent with other studies that have linked *RYK* expression to drug sensitivity or resistance, suggesting that *RYK* plays a role in the response to chemotherapy. A possible mechanism is that knocking down *RYK* expression could alter Wnt signaling pathways that are involved in DNA damage response or cell death. These pathways are known to be critical in regulating cellular responses to DNA damage and may play a role in the sensitivity or resistance to TMZ treatment. Although further studies are needed to elucidate the mechanisms by which *RYK* affects TMZ response or how TMZ treatments modulate *RYK* expression, we show that increased levels of *RYK* expression have been associated with increased viability when treated with TMZ in two cell line model systems, suggesting a portion of inter-individual variability may be mediated by *RYK* expression via a common pathway.

However, testing multiple cell lines with the AA genotype would strengthen the evidence of the association between the genotype and the phenotype, but SNP rs4470517 is not typically available in public cell line databases, and although we tested multiple glioma cell lines for the AA genotype, we only found one readily available; there would likely be others in private cell line data. Additionally, while HEK cells are commonly used for transfection experiments, it is important to acknowledge that the results may not necessarily reflect what happens in glioma cells. It is worth noting that HEK cells can still provide useful information regarding the function and regulation of genes of interest, which can inform further investigation in glioma cells or other relevant models. Regarding the *RYK* knockdown and its effect on cell viability, it is possible that the reduced effect observed in the glioma cell line may be due to differences in the genetic and epigenetic backgrounds of glioma cells compared to HEK cells. It may be necessary to explore other factors that contribute to TMZ resistance in glioma to fully understand the implications of *RYK* regulation in glioma.

While we recognize and acknowledge these limitations of our study, we believe that our findings provide valuable insights into the role of this *RYK* genotype in TMZ sensitivity, and attempt to interpret our results with caution. We hope that our work will inspire further investigation in this area and may provide further evidence of the potential utility of *RYK* as a predictive biomarker for TMZ response.

*RYK* is a pseudo-kinase family member, a receptor or coreceptor that can bind to multiple Wnt ligands (glycoproteins that are cysteine-rich and highly hydrophobic) and has been linked to several Wnt signaling pathways (WNT5A), predominately the non-canonical pathway [29,30,31]. Alterations and dysregulation of the Wnt pathways result in cancers and other disease conditions. The Wnt system is divided into canonical and non-canonical pathways, the canonical Wnt pathway mainly regulates cell differentiation and proliferation, whereas the non-canonical Wnt pathway regulates cell polarity, adhesion, and motility [32,33]. Recent studies have established a role for the Wnt pathways, having been positively correlated to the continued development, progression, and invasiveness of gliomas with a poor patient prognosis [34,35,36,37,38,39,40]. Other studies have also found that the Wnt signaling pathways are associated with regulation of TMZ response [41,42,43,44,45,46,47,48,49,50]. Wnt signaling regulation has been associated with multiple anticancer drug responses in other cancers as well [51,52,53,54]. These findings point to an important pharmacological mechanistic role for this pathway in glioma malignancy [29,34,35,55].

*RYK* is known to be highly expressed in many tumors and associated with advanced stages, metastasis, increased migration, and invasion [56,57,58,59,60,61,62]. *RYK* expression has also been associated with tyrosine kinase inhibitor drug tolerance and resistance in lung cancers, and BRAF inhibitor resistance in melanomas [63,64]. Additionally, with the inhibition of *RYK* with anti-*RYK* antibodies in mice with hematological cancers, the mice were sensitized to fluorouracil (5-FU) treatment [65,66]. *RYK* has been shown to regulate Wnt signaling, which in turn can induce cellular quiescence, where TMZ and many conventional chemotherapeutic drugs are dependent on increased proliferation in malignant cells. *RYK* is shown to have an important role in the capacity to better manage the stress and repair response of the cells through Wnt5a-induced suppression of reactive oxygen species [67]. While cell line model systems are helpful in testing drug response, their limitations due to the lack of representation of the complex cellular microenvironment of human tumors prevent us from distinguishing whether the response is modeling efficacy or toxicity. Therefore, additional model systems such as spheroids and animal models are necessary. Spheroids can mimic the tumor microenvironment, providing more accurate assessment of drug response [68,69], while animal models can provide preclinical data on drug efficacy and toxicity in vivo [70,71]. Incorporating these models can provide more comprehensive data on drug response and inform clinical trial design. Our findings suggest efficacy and provide valuable insights for translating our results into these additional model systems and ultimately human studies.

Previously, rs4470517 has only been reported as a potential genetic risk factor for oral clefts birth defects [72]. Predictive tools, databases, and limited evidence used to assess the functional impact of the non-coding genetic variant suggest that rs4470517 may exert its influence on *RYK* expression through an alternate splice region variant resulting in a consequential retained intron or nonsense mediated decay of the transcript. Other regulatory network interactions may also be implicated. A separate LD analysis also identified an SNP rs56088702, not flagged in the GWAS in low to moderate LD with rs4470517/rs4854617 (R^2^ = 0.37/0.42), which also occurs within a splicing region of *RYK* causing a splice polypyrimidine tract variant [73,74,75]. Together, these SNPs may contribute to a haplotype effect resulting in reduced splicing efficiency in the AA rs4470517 genotype variants resulting in decreased *RYK* expression and a more active sensitive response to TMZ with improved survival outcomes. Ethnic variation observed in rs4470517 MAFs may also contribute to the differences in ethnic incidence and response rate variability seen in glioma patients [76,77]. Our findings could have important implications for the development of personalized treatment strategies based on both genotype and gene expression levels. Patients with the AA variant may benefit more from TMZ treatment if their *RYK* expression is reduced, whereas those with the GG genotype may require alternative treatment options.

Our study aimed to evaluate the expression status of *RYK* as a predictor of time-to-event outcomes such as OS and PFS in glioma patients. Our analysis revealed that low *RYK* expression status was independently significantly associated with better survival outcomes in *IDH* mutants in adult-type diffuse glioma patients. However, in the GSE107850 dataset with a smaller sample size and lacking information on *MGMT* methylation status, the findings only suggest a trend indicating that low *RYK* expression status may be associated with improved survival outcomes in adult-type diffuse glioma patients with *IDH* mutations.

In the case of *IDH* wildtype GBM patients, our analysis revealed that only *MGMT* methylation status was significantly associated with survival outcomes. *MGMT* methylation status is considered both a prognostic and predictive biomarker for GBM patients undergoing treatment with TMZ. Studies have shown that *MGMT* methylation correlates with better response to treatment and improved overall survival [78]. *MGMT*’s role in DNA repair and TMZ’s ability to cause DNA damage likely contribute to its significant impact on GBM patient survival. However, further analysis showed a potential benefit of *RYK* expression in GBM patients with *MGMT* methylation. This suggests that the combination of *RYK* expression status and *MGMT* methylation may serve as a promising prognostic biomarker for GBMs and warrants further investigation.

Nonetheless, our current research has some limitations. Our findings from the TCGA *IDH* wildtype GBM dataset suggest that given the small sample sizes that met our inclusion criteria, further studies are necessary to accurately determine the significance of our findings. Although *IDH* mutations and *MGMT* methylation status are both molecular markers used in the diagnosis and treatment of gliomas, it is important to acknowledge that discrepancies in reported molecular markers for all patients may be due to changes in testing and classification guidelines over past years with older studies not reporting *IDH* mutation and *MGMT* methylation status as part of the analysis.

Our study has highlighted the intricate relationship between molecular biomarkers such as *RYK* expression and *MGMT* methylation and *IDH* mutant statuses, as well as their impact on patient survival outcomes in different types of gliomas. Despite the complexity of these interactions, our findings demonstrate the potential prognostic value of *RYK* expression status in glioma patients. Further research is necessary to fully understand the implications of these findings and to translate them into clinical practice.

## 4. Materials and Methods

### 4.1. Cell Lines Culture and Genetic Data

To screen 44 anticancer agents at six different doses each, for cell viability [21], 680 immortalized LCLs from the 1000 Genomes from nine geographically representative and ethnically diverse populations were used. Genotype data (which included the genotypes for rs4470517 and rs4854617 SNPs) from the Illumina HumanOmni2.5 platform were downloaded and processed as described in Abdo et al. [79]. Baseline RNA-Seq expression data are publicly available from the Geuvadis project for 272 of the LCLs in this study [80].

U87MG, U138MG, U373MG and T98G human glioma cell lines were obtained from the UNC Lineberger Comprehensive Cancer Center’s Tissue Culture Facility, with negative mycoplasma contamination test and STR profiling of cell lines originally from frozen stocks obtained from the American Type Culture Collection (ATCC, Manassas, VA, USA). Cells were kept frozen in liquid nitrogen. After thawing, glioma cell lines were cultured in Eagle’s minimum essential medium (Life Technologies, Carlsbad, CA, USA) supplemented with 10% (*v*/*v*) fetal bovine serum (Gibco, Waltham, MA, USA) and 1% (*v*/*v*) penicillin/streptomycin (Life Technologies, CA, USA). LCLs were cultured in RPMI growth media consisting of RPMI 1640 plus L-glutamine (Mediatech, Hsinchu, Taiwan) with 10% fetal bovine serum (Gibco) and 1% penicillin/streptomycin (Invitrogen, Carlsbad, CA, USA). Cells were maintained in culture at 37 °C under a humidified atmosphere containing 5% CO_2_. Each cell line was seeded onto 384-well plates with approximately 5000 cells/well. Cell count and viability were measured using Trypan dye exclusion.

### 4.2. TMZ Dose Response

TMZ (Sigma Aldrich, St. Louis, MO, USA, 25 mg) was dissolved in dimethyl sulfoxide (DMSO) for in vitro experiments. The concentrations of DMSO were 0.06% 0.16%, 0.33%, 0.5%, 0.67%, 1%, and 1.67%. Serial dilutions were made and TMZ was assayed at a range of concentrations in culture medium to final concentrations of 0, 0.25, 0.5, 0.75, 1, and 1.5 mM with the exception of assaying the glioma cells, which included an additional concentration of 2.5 mM. TMZ viability data were available for all LCLs (*n* = 680) from the anticancer agent screen at all six doses [21]. Each plate included a control for background signals, 50% and 10% DMSO, and drug vehicle. Each cell line was incubated for 48, 72, and 96 h for all the tested concentrations. AlamarBlue (Life Technologies, CA, USA) and a Tecan F200 plate reader with measure fluorescence intensity at EX535 nm and EM595 nm and iControl software (Version 1.6) were used to measure the resulting raw fluorescence units (RFUs). RFUs are proportional to the concentration of living cells in each well and used for viability measurements.

### 4.3. SNP Genotyping

DNA was extracted from U87MG, U138MG, U373MG and T98G cultured cells, using a QIAamp DNA Mini Kit (Qiagen, Germantown, MD, USA) in accordance with the manufacturer’s instructions. DNA quantity and quality were evaluated by nanodrop spectrophotometer (NanoDrop Technologies Inc., Wilmington, DE, USA). All samples were genotyped for rs4470517 (G > A) by TaqMan allelic discrimination assay (Life Technologies, CA, USA), and carried out according to the manufacturer’s instructions using a CFX384 Real-Time PCR detection system (Bio-Rad Laboratories, Hercules, CA, USA) with predesigned TaqMan SNP Genotyping Assay, human (assay ID:C__30400301_10) (Life Technologies, CA, USA). Approximately 25 ng of gDNA was loaded per well onto 384-well plates with TaqMan Genotyping Master Mix (Life Technologies, CA, USA) containing VIC and FAM reporter dyes. DNA from the 1000 Genomes Project with known SNPs served as the controls while performing genotyping for cancer cell lines. No-template reactions were included as controls in each assay run.

### 4.4. RYK mRNA Expression

Total RNA was isolated from ~1 × 10^6^ cells using the RNeasy kit (Qiagen, MD, USA). Reverse transcription-PCR was performed using the Verso cDNA kit (Life Technologies, CA, USA) according to the manufacturer’s instructions. cDNA was amplified by real-time PCR using a CFX384 Real-Time PCR detection system (Bio-Rad Laboratories, CA, USA) to determine *RYK* expression before and after TMZ treatment. The total reaction volume of 10 μL contained 50–100 ng cDNA. Each sample was analyzed using the predesigned TaqMan Gene Expression Assay *RYK* (Hs00243196_m1), as well as an endogenous control *GADPH* (Hs02786624_g1) (Life Technologies, CA, USA). Ct-values for all samples were determined using FPK-PCR [81]. *RYK* mRNA levels were calculated by the comparative CT method using GAPDH as endogenous housekeeping genes. Each cell sample was measured in triplicate. The means (standard error) for each cell line were averaged for each group.

### 4.5. RYK Overexpression and siRNA Knockdown

The *RYK* expression clone is a lentiviral-based plasmid under a CMV promoter used for transfection. The plasmid has a puromycin resistance gene for stable selection. Transfections were performed using the Lipofectamine 3000 kit (Life Technologies, CA, USA) according to the manufacturer’s instructions. HEK293 and U138MG cells were infected with recombinant lentivirus-transducing units. Stable selection of infected cells was maintained with 1 μg/mL of puromycin (Sigma, St. Louis, MO, USA) for 7 days before they were seeded for growth curve experiments and protein/mRNA analysis to determine overexpression. An empty plasmid was used as a control. *RYK* overexpression was confirmed by Western blot.

HEK293 and T98G cells were transfected with Silencer Select siRNAs Predesigned small interfering RNA (siRNA) targeting human *RYK* (ID#: s12390 and s12392) and Silencer Select Negative Control #1 and Silencer Select GAPDH positive control (Life Technologies, CA, USA) at a final concentration of 50 nM using Lipofectamine RNAiMAX (Life Technologies, CA, USA) according to the manufacturer’s instructions. Following a 48 h incubation time, cells were harvested post-transfection for analysis of protein and RNA levels. *RYK* knockdown was confirmed by Western blot analysis.

### 4.6. Western Blot Analysis

Cells were collected, rinsed in cold PBS, and lysed in RIPA lysis and extraction buffer reagents with Protease Inhibitor Cocktail kit (Life Technologies, CA, USA) on ice. Protein concentration was determined using BCA Protein Assay reagents (Life Technologies, CA, USA), according to the manufacturer’s instructions. Proteins (25 µg) were separated by electrophoresis in SDS-PAGE Tris/glycine gels (Bio-Rad Laboratories, CA, USA) and blotted onto a nitrocellulose membrane (Life Technologies, CA, USA). Samples were incubated in blocking buffer before the addition of the 1:1000 diluted primary antibody—a rabbit anti-*RYK* antibody (Proteintech, Rosemont, IL, USA). A polyclonal anti-GAPDH (Life Technologies, CA, USA) at a 1:1000 dilution was used as a control. Alexa Fluor secondary antibodies (Goat anti-rabbit Alexa fluor 700 A21038, Goat antirabbit Alexa fluor 800 A32735) were used as needed at a 1:1000 dilution. Detection was performed on a LiCOR Odyssey to visualize protein bands (LI-COR Biosciences, Lincoln, NE, USA). Protein density was calculated using ImageJ and Fiji software [82,83]. The polyclonal anti-GAPDH antibody used in our experiment has the potential to result in non-specific binding, leading to the appearance of additional visible bands in the GAPDH Western blot (Appendix A). The most prominent band observed aligns with the formation of dimers, trimers, or multimers. Increased boiling time during sample preparation has been shown to reduce the formation of multimers and potentially decrease non-specific binding.

### 4.7. Clinical Datasets

Univariate and multivariate Cox regression analyses were performed using glioma datasets from two main data sources. First, the expression and clinical data of GBM and LGG datasets from TCGA were downloaded from University of California Santa Cruz (UCSC) Xena browser and the cBioPortal [84,85,86]. The samples with missing data on survival and *RYK* gene expression were excluded in this study, and 530 LGG and 171 GBM glioma patients were finally obtained in the TCGA dataset. Second, available *RYK* gene expression profiles (probes: 216976_s_at_3, ILMN_1769671) and clinical survival information were collected from the GEO database [56] (http://www.ncbi.nlm.nih.gov/geo/, accessed on 31 August 2022.) from two independent glioma clinical trial cohorts. The GSE107850 dataset [87] included 195 samples of LGG patients’ samples treated with RT or TMZ, and the GSE7696 dataset [88,89] included 80 samples of GBM patients’ samples treated with RT or TMZ and RT. We categorized patient cohorts into three strata based on their *RYK* gene expression values: low (1–33 percentile), medium (34–66 percentile), and high (67–100 percentile).

Since rs4470517 is a novel association in a non-coding intron region, no genotype information for this SNP was available for patients due to lack of whole genome sequencing.

### 4.8. Statistical Analysis

All experiments were performed in triplicate unless otherwise indicated. All results are expressed as the mean ± standard error. All statistical analyses were performed using R version 4.0.3 and RStudio. ANOVA, Student’s *t*-test, and post hoc Tukey, as appropriate, were used to analyze the significance of differences in gene expression levels. Pearson correlation analysis was used to estimate the relationship between the baseline expression level of *RYK* and TMZ AUC. The R package “roc” and “PharmacoGx” were used to calculate the area under the curve (AUC) and significance [90]. DataGraph (Version 4.6) (Visual Data Tools, Inc. Chapel Hill, NC, USA. https://www.visualdatatools.com/ Accessed on 15 February 2022.) was used to produce the figures. Kaplan–Meier and Cox analyses were performed using the packages “survival” and “survminer” in RStudio and patients were divided into trichotomized groups of *RYK* expression. Kaplan–Meier curves were compared using the log-rank test and hazard ratio (HR) using the Cox proportional-hazards model. Kaplan–Meier plots were drawn to visualize survival differences using the “survivalROC” and “ggplot2” R packages. We performed univariate Cox regression to evaluate the association between each predictor variable and time-to-event outcomes. We then employed multivariate Cox regression to account for potential confounding effects of each covariate and to analyze the relationship between predictors and time-to-event outcomes. By using both univariate and multivariate analyses, we were able to gain a better understanding of the independent and joint effects of *RYK* expression, *IDH* mutation status, and *MGMT* methylation status on survival outcomes. All *p*-values were two-sided, and differences were defined as statistically significant for *p*-values < 0.05.

## 5. Conclusions

According to our study, we suggest that *RYK* may provide an additional marker to guide glioma therapy. We found that the rs4470517 SNP in *RYK* is associated with the regulation of *RYK* gene expression and TMZ response in cell lines, and we demonstrated that overexpression and knockdown of *RYK* impacts TMZ response as predicted. Additional studies are needed to clarify by what mechanism *RYK* works to exert its influence on TMZ response, and additional sequencing data are needed to further implicate the SNPs’ direct association with gene expression. Likewise, as in any model system, our results need to be validated in human studies to further translate our findings into a clinical setting. These findings have valuable implications in understanding a possible role that *RYK* has in glioma pathogenesis and the possibility of new targeted therapies.

Our results indicate that *RYK* expression could potentially function as a significant predictor or prognostic factor for the response and survival of glioma patients. As the field of glioma research continues to evolve, staying up to date on current guidelines is essential, as it can impact the interpretation of study results.

## Figures and Tables

**Figure 1 pharmaceuticals-16-00726-f001:**
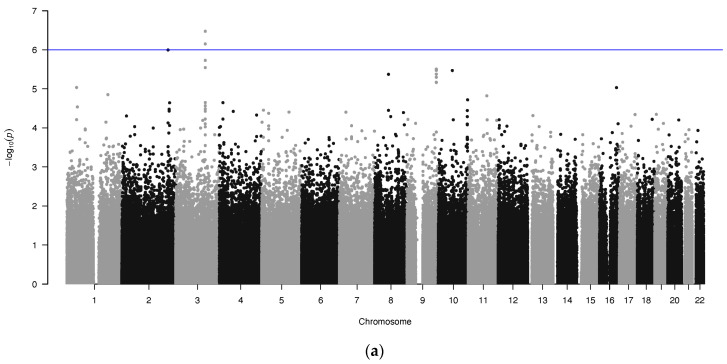
(**a**) Manhattan plot −log10 *p*-values for the GWAS for the drug temozolomide. The solid line indicates the threshold for suggestive significance. The peak on chromosome 3 has 2 SNPs surpassing the suggestive significance; (**b**) locus zoom plots showing the region surrounding rs4470517 and rs4854617 SNPs on chromosome 3 for associations with temozolomide. The lead/reference SNPs rs4470517 (blue) and rs4854617 (red) are shown as diamonds. For all other non-lead SNPs shown as circles and triangles, their color and shape are matched to the lead SNP with which it is in the highest linkage disequilibrium (LD).

**Figure 2 pharmaceuticals-16-00726-f002:**
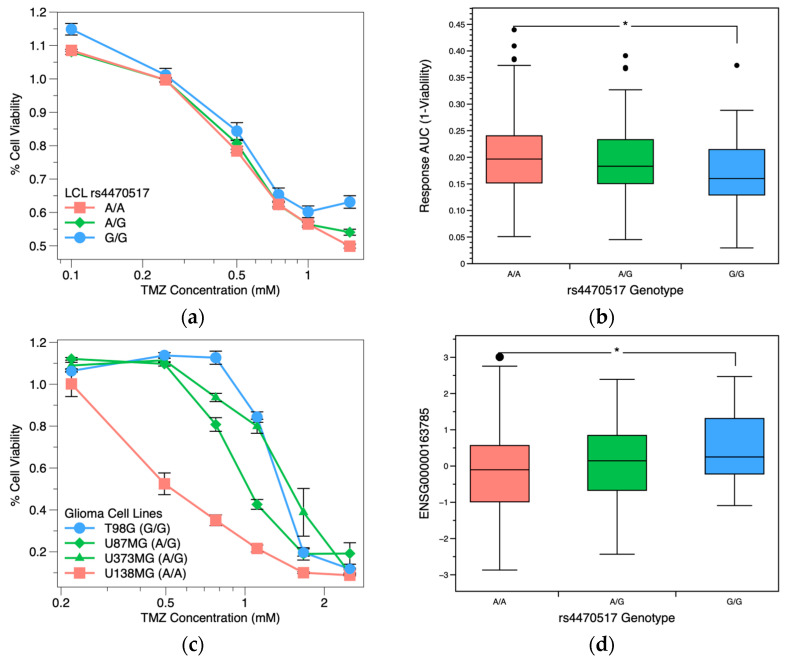
Associations for *RYK* response data, gene expression in 1000 Genome LCLs stratified by genotype rs4470517. Concentrations are on log scale. Error bars represent the standard error of the mean. (**a**) LCL’s dose-response profiles for temozolomide stratified by rs4470517 genotype. Each genotype is represented by multiple cell lines, with a total of *n* = 648 samples: GG—32, GA—184, AA—432. (**b**) Box plot of the AUC area (response area = 1—viability) for TMZ by genotype. A higher AUC indicates higher drug response to TMZ treatment. (**c**) Glioma cell lines’ dose-response profiles for temozolomide stratified by rs4470517 genotype. U138MG cells have an AA genotype, low cell viability, and an IC50 of 0.57 mM TMZ. T98G cells have a GG genotype, high cell viability, and an IC50 of 1.44 mM TMZ. U87MG and U373MG cells have a heterozygous GA genotype and have intermediate cell viability and an IC50 of 1.08 mM and 1.51 mM TMZ, respectively. (**d**) Box plot of *RYK* transcript levels (ENSG00000163785) by genotype. * *p*-value < 0.05. Black circles in box plot represent outliers.

**Figure 3 pharmaceuticals-16-00726-f003:**
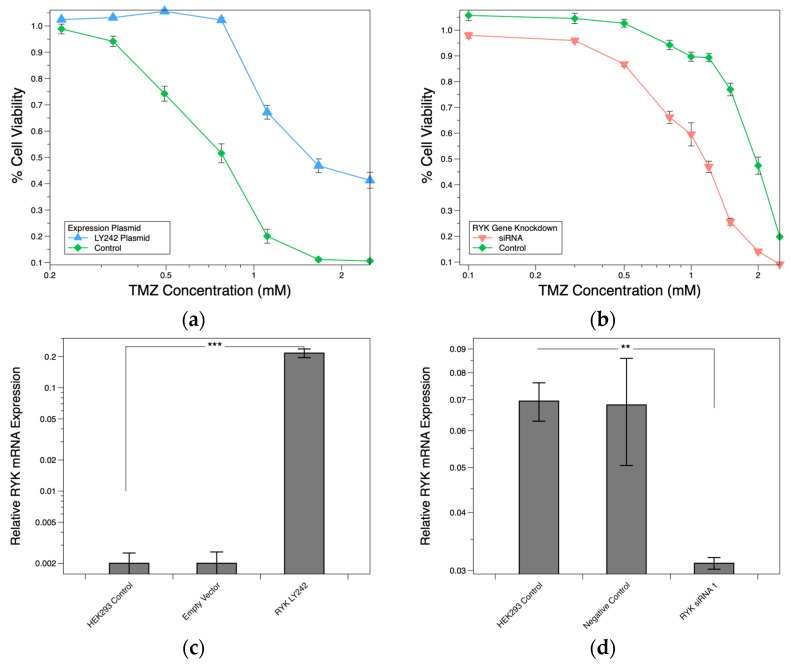
HEK293 cell line *RYK* overexpression and knockdown. (**a**) *RYK* overexpression TMZ dose-response profile. (**b**) *RYK* siRNA knockdown TMZ dose-response profile. Concentrations are on the log10 scale. Error bars represent the standard error of the mean. (**c**) *RT-qPCR RYK* overexpression (*n* = 5), *RYK* mRNA levels calculated by the comparative CT method. (**d**) *RT-qPCR RYK* siRNA knockdown (*n* = 5), *RYK* mRNA levels calculated by the comparative CT method. Each cell sample was measured in triplicate. The means (standard error) for each cell line were averaged. ** *p*-value < 0.01, *** *p*-value < 0.001.

**Figure 4 pharmaceuticals-16-00726-f004:**
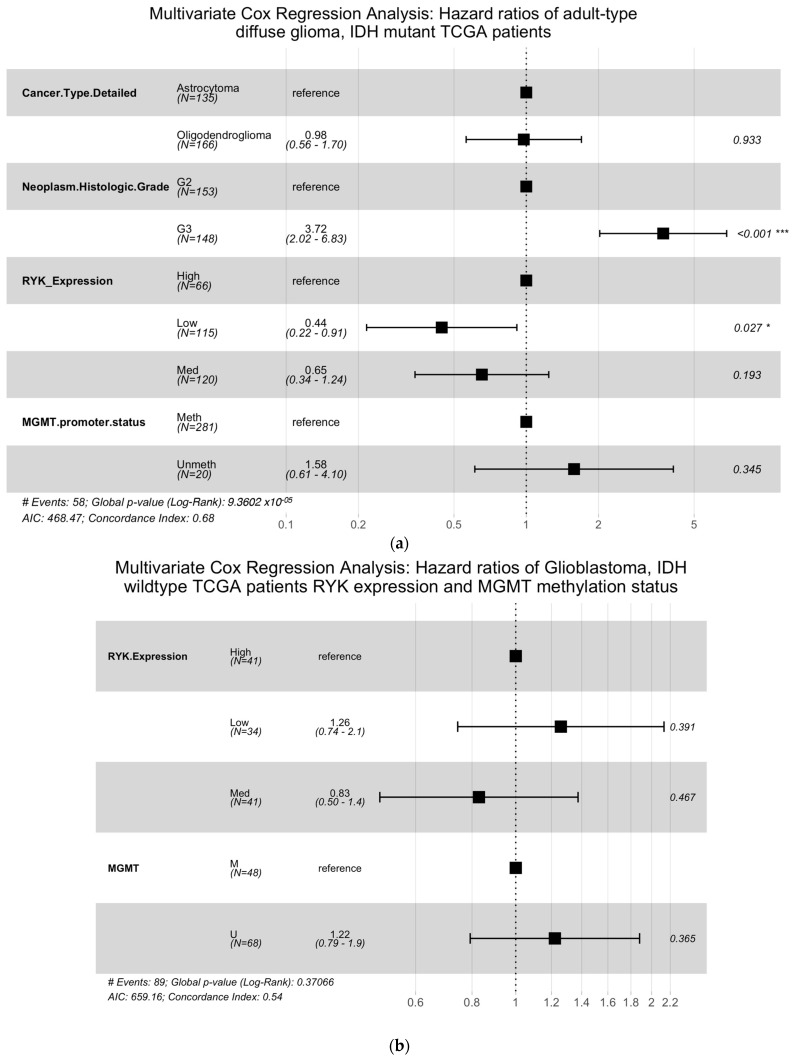
Multivariate Cox proportional hazards regression analysis. (**a**) Forest plot showing the HR and CI for OS of the TCGA *IDH* mutant in adult-type diffuse glioma patient cohort. (**b**) Forest plot showing the HR and CI of OS of the *IDH* wildtype GBM TCGA patient cohort. * *p*-value < 0.05, *** *p*-value < 0.001.

**Figure 5 pharmaceuticals-16-00726-f005:**
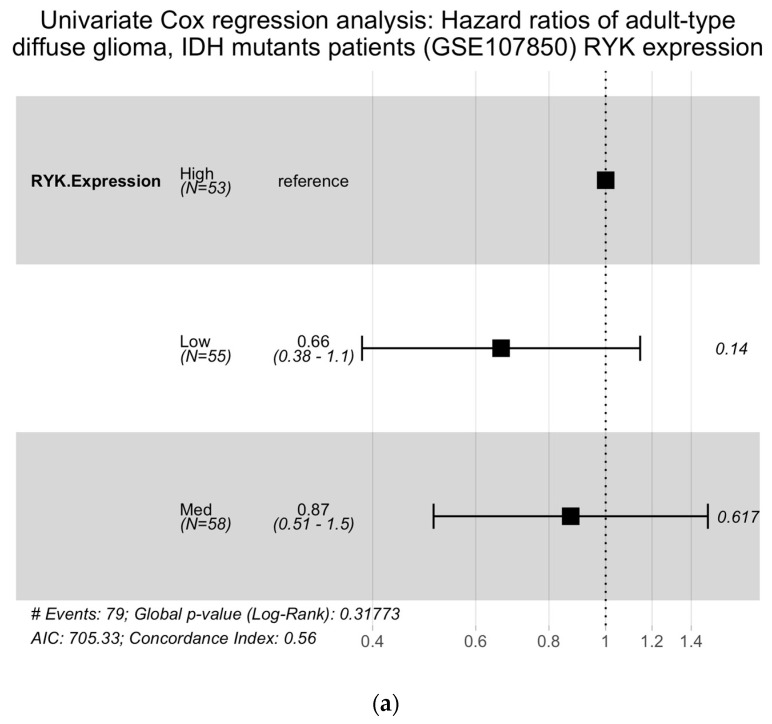
Multivariate Cox proportional hazards regression analysis. (**a**) Forest plot showing the HR and CI for PFS of the GSE107850 *IDH* mutant in adult-type diffuse glioma patient cohort. (**b**) Forest plot showing the HR and CI of OS of the GSE7696 *IDH* wildtype GBM patient cohort. *** *p*-value < 0.001.

## Data Availability

Data available within the article and its Appendix A

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
