# Peer review of "RYK Gene Expression Associated with Drug Response Variation of Temozolomide and Clinical Outcomes in Glioma Patients"

_pharmaceuticals, 2023, doi:10.3390/ph16050726_

Round 1

Reviewer 1 Report (Previous Reviewer 1)

in my opinion the manuscript is suitable for publication 

Author Response

Reviewer 2 Report (Previous Reviewer 2)

Overall, the authors have made efforts to address many of my concerns. Particularly, I appreciate the effort to include data regarding IDH and MGMT status within the data gathered from TCGA. However, there are still major issues in this study, some of which are new based on revisions provided.

- Incorrect nomenclature is still commonly used throughout the manuscript regarding glioma classification. Within the introduction, the terms "diffuse astroctyoma", "anaplastic astrocytoma", and "anaplastic oligodendroglioma" are mentioned; however, these diagnoses are no longer appropriate and are even discouraged. Further, it is incorrect to say "Astrocytic tumor, IDH-mutant, WHO grade 4"; instead, it is better classified simply as "Astrocytoma, IDH-mutant, WHO grade 4". These points might seem a simple matter of semantics; however, using correct terminology gives significant credence that the authors are knowledgeable about the topic they are writing about.

- The term "Low-grade glioma (LGG)" includes not only the diffusely infiltrating tumors (i.e., Astroctyoma, IDH-mutant, WHO grade 2; and Oligodendroglioma, IDH-mutant & 1p/19q codeleted, WHO grade 2); but also includes additional grade 2 gliomas (I.e., PXA) as well as a variety of grade 1 gliomas (notably, pilocytic astrocytoma, amongst others). These circumscribed LGGs generally do NOT harbor IDH mutations. The authors should clarify this in the text. In the context of this particular paper, the authors refer to diffusely infiltrating LGG and do not seem to include the circumscribed astrocytic gliomas. 

- It is unclear to this reviewer why information about glioma epidemiology was included (lines 72-75). This seems outside of the scope of this study.

- The authors should include that gross total surgical resection, when possible, is also part of the standard care for glioma (lines 75-77), especially high-grade glioma.

- The authors should comment more directly, probably in the results but definitely at least in the discussion, the limitation of having only found one primary glioma cell line with the AA genotype, as discussed in their response to my concern on this point. This is true of their response regarding HEK cell line data as well. These are important limitations which should be included in the discussion as elements of limitations of this study. 

- In section 2.6, there are again references to out of date diagnoses, especially "Oligoastroctyoma", which has been discouraged even since the 4th Edition of the CNS WHO in 2016. While a valuable dataset, data gathered from TCGA needs to be carefully vetted and assessed in order to be consistent with current knowledge of these tumors. Similarly, items with incomplete and incorrect data in these large datasets should be removed or separated when the presented information is inconsistent with current understanding. It is also incorrect to group "Anaplastic astrocytoma" and "anaplastic oligodendroglioma", both WHO grade 3 tumors, within the LGG category (even though the data was probably retrieved from a TCGA dataset labeled broadly as "LGG"). Grade 3 tumors are not "low-grade", and thus, should not be evaluated as such. Similarly, the results state "Amongst patients with LGGs, 97% has tumors classified as ...". What were the remaining 3% classified as? These issues affect the univariate and multivariate analyses presented thereafter.

- Within the same TCGA analysis, "94% of patients with GBMs had tumors that were IDH-wildtype...". Current classification schemes define glioblastoma as IDH-wildtype. The remaining 6% of tumors with IDH mutation should be excluded from this group. These issues also affect the univariate and multivariate analyses presented thereafter.

I appreciate the efforts to include data related to IDH and MGMT status. However, there are still significant issues related to how the authors collected and organized the data, thereby limiting the reliability and general interest of the presented results. This is an issue with any study, particularly older ones and any study that does not utilize current classification guidelines, because it limits comparison between studies and potentially misrepresents the true nature of this diverse group of neoplasms. While older studies cannot be changed, current and ongoing studies can and should be performed in line with current thought and understanding. 

- In a new section of the discussion, the authors assert that "We observed that individuals with the AA variant have significantly decreased expression ...". As far as I can tell, all of the data related to AA/AG/GG genotypes in relation to TMZ sensitivity was in relation to cell lines, not "individuals". The tone is changed within the same paragraph to refer to "groups" rather than "individuals"; however, the authors should be more clear that this data was primarily gathered from cell culture experiments. 

Author Response

Reviewer 3 Report (Previous Reviewer 3)

Ref: ID: pharmaceuticals-2341653

The manuscript entitled: “RYK Gene Expression Associated with Drug Response Variation of Temozolomide and Clinical Outcomes in Glioma Patients” by Gonzalez et al., is a research article suggesting that SNP rs4470517 in the RYK (Receptor-Like Kinase) gene, is associated with RYK expression and RYK expression status was also found to be a leading and influential predictor of TMZ response and survival outcomes for glioma patients. This is an interesting study, that fits within the scope of the journal. The authors responded adequately to most of my comments and suggestions. I believe the revisions have significantly improved the quality of the manuscript. I have 2 additional minor comments before the acceptance for publication in the Pharmaceuticals  journal.

1) Please make figure 5 bigger and improve the visibility to the readers.

2) Suppementary Figure S5: The GAPDH western blot has other visible bands as well, therefore the authors need to elaborate over the specificity of this experiment.

Round 2

Reviewer 2 Report (Previous Reviewer 2)

I appreciate the concerted efforts by the authors to address my concerns. Overall, my comments have been adequately addressed and the manuscript in its current form is suitable for publication. 

This manuscript is a resubmission of an earlier submission. The following is a list of the peer review reports and author responses from that submission.

Round 1

Reviewer 1 Report

Thank you for submitting you valuable work. Neuro-oncology is a rapidly evolving field and since the new classification of gliomas much progress was made in understanding this type of tumors.

The article RYK Gene Expression Associated with Drug Response Varia- 2 tion of Temozolomide and Clinical Outcomes in Glioma Patients is a well written article with a possible impact in every day practice.  

In vitro studies reported in the article are well conducted.

May I suggest some changes from the clinician point of view.

-          Regarding figure 4 I believe the data may be misleading considering the molecular differences and prognostic between Low grade gliomas and glioblastoma. Date presented after in the text show different outcome, and personal I will keep just the Kaplan Meier curve subdivided to GBM and  LGG.

-          Statistical analysis that show thar RYK expression may modulate response to TMG and TMZ may modulate TYK expression may be analyzed using multivariat analysis considering the influence of IDH and MGMT metilation.

Reviewer 2 Report

In the manuscript by Gonzalex et al., the authors build off another study which demonstrated possible genetic determinants for temozolomide (TMZ) response. In particular, unique SNP within the RYK gene was identified and studied more extensively in this manuscript. The authors demonstrate using in vitro LCL and glioma cell lines that the AA genotype is associated with decreased RYK gene expression and increased TMZ response (cell viability assay). The authors then perform RYK overexpression and knockdown studies in lines with AA genotype or GA/GG genotype, respectively, to show reversal of effects in TMZ response. Finally, the authors utilize data from large public databases (TCGA, etc.) to assess RYK expression and infer possible TMZ susceptibility and associated survival in patients with glioblastoma (GBM) and low-grade glioma (LGG). While there is potential merit to this study, there are several major methodologic and scientific issues which must be addressed:

- Western blots only show one lane for each condition, whereas the methods suggest all experiments were done in triplicate. Results from all biologic replicates should be shown in the western blots, and the entirety of the western blots should be made available for review (not just the cropped images). 

- The 2016 WHO classification of CNS tumors is cited in the introduction (line 55) and corresponding old terminology is used throughout the manuscript (including roman numerals for grading). The most current 2021 WHO classification should be utilized, including current diagnostic nomenclature and terminology.

- On lines 162-163, the authors state "Having previously observed that TMZ treatment reduced RYK expression in the cell line with the AA genotype...". This is an incorrect interpretation of the data presented. They do not show that TMZ treatment has an effect on RYK expression. Instead, they show that RYK expression influences TMZ responsiveness.

- For glioma cell line studies, only one cell line was identified as having the AA genotype. Additional cell lines should be examined to find additional lines with the AA genotype to strengthen the assertion that this particular genotype influences glioma responsiveness to TMZ. Similarly, while RYK knockdown in HEK cells led to decreased cell viability in response to TMZ, the effect was minimal in the glioma cell line, again raising the question as to how relevant these data are in glioma.

- Finally, in figures 4 and 5, the authors use TCGA and other public databases to assess RYK expression in relation to overall patient survival. There are several issues with these data. First, the authors do not take into account MGMT promoter methylation status or IDH gene status, both of which are independently associated with better outcome. It could be that the high RYK expression tumors are all IDH-wildtype glioblastoma that lack MGMT promoter methylation. Similarly, GBM and LGG should not be grouped - these are completely different tumors and it is like comparing apples to oranges. The authors should also specify whether the tumors they have designated as "GBM" are IDH-wildtype (in previous iterations of the WHO, as cited in this paper, grade 4 IDH-mutant astrocytoma was formerly referred to as IDH-mutant glioblastoma). Any of these factors could influence their survival data independently of RYK expression level. Indeed, the authors state that RYK expression represents "a distinct mechanism of TMZ related responses different from causes of IDH mutations and MGMT methylation status" (line 309-310) without, as discussed above, actually showing this.

- It is unclear how the three-tier RYK expression cutoff (low, medium, high) was determined in Figures 4 & 5? 

Other minor points:

- The authors state that molecular markers such as MGMT and IDH "only affect a small portion of tumors" (line 61). These are two of the most important clinical and pathological markers in adult-type infiltrating tumors and affect a large number of patients, with as many as 50% of glioblastoma having methylation of the MGMT promoter in some studies. 

- The relative RYK mRNA gene expression differences in the LCL experiments, while statistically significant, are relatively small (Figure 2c). Corresponding Western blot experiments with quantification would be nice to see. Also, was RYK gene expression analysis performed on the glioma cells lines corresponding to Figure 2d? If not, why not?

- For some reason, the data in figure 3 is split between two pages (panels a-b on page 5 and panels c-f on page 6), making the data difficult to interpret. Also, it makes more sense to show the gene expression and protein data first, followed by the cell viability assays. 

- Lines 258-267 of the discussion are just a reiteration of the results with real no discussion integrating this data into the current literature.

- There are several typographical/grammatical errors throughout the manuscript (line 32, 48, 73, 188

Reviewer 3 Report

Ref: ID: Pharmaceuticals-2274412

The manuscript entitled: “RYK Gene Expression Associated with Drug Response Variation of Temozolomide and Clinical Outcomes in Glioma Patients” by Gonzalez et al., is a research article suggesting that SNP rs4470517 in RYK (Receptor-Like Kinase) gene, is associated with RYK expression and RYK expression status could be an influential predictor of TMZ response and survival outcomes for glioma patients. This is an interesting study, that fits within the scope of the journal. The manuscript needs some major revisions before acceptance for publication in the Pharmaceuticals  journal and please find below comments/suggestions and revisions that will further help the authors improve the current version of the manuscript.

Further comments:

Abstract:

-Lines 31-32: please rephrase the sentences to be more consistent to the publicly available bioinformatic data that have been used.

Introduction:

-Line 44: Usually the high grade gliomas are considered the High-grade [WHO grades III–IV] gliomas. Also, according to the WHO 2021 we currently have a new classification system not only according to the histological grades but also according to other criteria. Please elaborate and enrich the introduction according to these new updates.

-Line 80: please mention briefly which other drug treatments are currently being used for gliomas, other than TMZ and justify the choice of this drug for chemoresistance in your study.

Results:

- GWAS Analysis Figure 1a,b: please specify whether these results are original or taken from a previous publication(not very clear the way was written in the text).

-Line 172: “RYK gene expression profiles after 72 hrs…”: please explain why there was a choice for treatment after 72h Temozolamide not a time point 24h, 48h, 72h, d4 etc.

-Line 149-150: please mention whether the overexpression was transient in the control and glioma cell line. Also the authors would be better to choose more glioma cell lines for transfection in addition to the one non-glioma. Please elaborate.

-line 162-164: “Having previously observed that TMZ treatment reduced RYK expression in the cell lines with the AA genotype, we hypothesized that RYK knockdowns would result in a comparably, more sensitive phenotype as those with the AA genotype.”: please explain this result more in the discussion part.

-Figure 3: please upload as supplementary the full images for western blot and RT-PCR images/data. Also in this main figure 3, please include the molecular weight markers that were used in each panel.

-line 188-189: please rephrase this sentence since looks incomplete.

-line 204: still this sentence is incomplete.

-Kaplan-Meier curves of OS for GBM patients and PFS for LGG patients.”: please explain the choice of OS for GBM and PFS for LGG patients for your survival analysis. Authors would be better to choose to perform both types OS and PFS for all glioma patients and compare the results.  Also would be better to do further analysis for all glioma grades related to Disease-free survival (DFS) and metastasis-free survival (MFS).

Discussion:

-Line 295-296: “While cell line model systems are helpful in testing drug response, their limitations prevent us from distinguishing whether the response is modeling efficacy or toxicity.”: please explain here the importance of using additional model systems such as spheroids, mice models that will help overcome these limitations.

-Line 330: please correct the term "glioma cancer" to "glioma brain tumors".

Methods

-Cell lines: please include a statement about the cell line authentication and the mycoplasma contamination for all the cell lines that have been used.

-lines 419-422: please include the details for the dilutions of the primary and secondary antibodies for the western blot.